# Diagnostic performance of a Recombinant Polymerase Amplification Test—Lateral Flow (RPA-LF) for cutaneous leishmaniasis in an endemic setting of Colombia

**Alexandra Cossio**[1,2]*, **Jimena Jojoa**[1,2], **María del Mar Castro**[1,2], **Ruth Mabel Castillo**[1,2], **Lyda Osorio**[3], **Thomas R. Shelite**[4], **Nancy Gore Saravia**[1,2], **Peter C. Melby**[4], **Bruno L. Travi**[4]*

**1** Centro Internacional de Entrenamiento e Investigaciones Médicas, CIDEIM, Cali, Colombia, **2** Universidad Icesi, Cali, Colombia, **3** Universidad del Valle, Cali, Colombia, **4** University of Texas Medical Branch, Galveston, Texas, United States of America

* acossio@cideim.org.co (AC); brltravi@UTMB.EDU (BLT)

**Data Availability Statement:** All relevant data are within the manuscript and its Supporting information files.

## Abstract

### Background

Control of cutaneous leishmaniasis by public health systems in the Americas relies on case identification and treatment. Point-of-care diagnostics that can be performed by health workers within or near affected communities could effectively bring the health system to the resource-limited sites providing early diagnosis and treatment, reducing morbidity and the burden of disease.

### Methodology/principal findings

A cross-sectional study was undertaken to evaluate the diagnostic test performance of Isothermal Recombinase Polymerase Amplification (RPA) targeting *Leishmania* kinetoplast DNA, coupled with a lateral flow (LF) immunochromatographic strip, in a field setting and a laboratory reference center. Minimally invasive swab and FTA filter paper samples were obtained by community health workers and highly trained technicians from ulcerated lesions of > 2 weeks' evolution from 118 patients' ≥ 2 years of age in the municipality of Tumaco, Nariño. Extracted DNA was processed by RPA-LF at a reference center or in a primary health facility in the field. Evaluation was based on a composite "gold standard" that included microscopy, culture, biopsy and real-time polymerase chain reaction detection of *Leishmania* 18S rDNA. Standard of care routine diagnostic tests were explored as comparators.

Sensitivity and specificity of RPA-LF in the reference lab scenario were 87% (95%CI 74–94) and 86% (95%CI 74–97), respectively. In the field scenario, the sensitivity was 75% (95%CI 65–84) and specificity 89% (95%CI 78–99). Positive likelihood ratios in both scenarios were higher than 6 while negative likelihood ratios ranged to 0.2–0.3 supporting the usefulness of RPA-LF to rule-in and potentially to rule-out infection.

**Funding:** This work was funded by "Ministerio de Ciencias Tecnología e Innovación -Minciencias", https://minciencias.gov.co/ [Grant 222972553501 to Alexandra Cossio], and was partially supported by the Center for Tropical Diseases, University of Texas Medical Branch. The funders had no role in study design, data collection and analysis, decision to publish, or preparation of the manuscript.

**Competing interests:** The authors have declared that no competing interests exist.

## Conclusions/significance

The low complexity requirements of RPA-LF combined with non-invasive sampling support the feasibility of its utilization by community health workers with the goal of strengthening the diagnostic capacity for cutaneous leishmaniasis in Colombia.

## Trial registration

ClinicalTrials.gov NCT04500873.

## Author summary

Limited access to diagnosis is a critical determinant of the "neglect" that defines the so-called Neglected Tropical Diseases (NTDs) including cutaneous leishmaniasis. Diagnostic tests that can be performed close to and involve the participation of the affected communities would improve access to treatment as well as diagnosis. Using non-invasive swab and filter paper samples obtained by Community Health Workers, we evaluated the diagnostic performance of an innovative and technically simple molecular test: Isothermal Recombinase Polymerase Amplification (RPA) to detect *Leishmania* DNA, coupled with a lateral flow (LF) strip to read the results with the naked eye. The RPA-LF test demonstrated high sensitivity and specificity and capacity to rule in or rule out a diagnosis of cutaneous leishmaniasis in both an endemic field setting and reference laboratory. The findings encourage the further optimization of the test format for Point-of-Care diagnosis by health personnel and rural health workers in endemic settings.

## Introduction

Cutaneous leishmaniasis (CL) is a recognized public health challenge in the Americas, with an average of 55,000 cases per year between 2001–2018 in 17 countries of the region [1]. Most cases (79.4%) have been acquired in and continue to occur in rural areas [2]. Colombia is second in reported cases in Latin America and one of the ten countries worldwide reporting the highest number of cases [3]. In 2018, 6273 new cases of CL were reported in Colombia [1], and Tumaco was among the most affected municipalities [4]. Leishmaniasis control efforts rely on case identification and treatment, both of which are challenging in rural areas where access to health services is often scarce. Hence, one of the goals of the Pan American Health Organization for control of leishmaniasis in the Americas 2017–2022 is to improve the opportunity and access to diagnosis, treatment, rehabilitation and adequate follow-up of leishmaniasis cases [5]. According to Colombian national guidelines, provision of treatment requires parasitological diagnosis [6]. Sensitive, specific and accessible diagnostic methods are needed to meet this goal, to prevent progression to mucosal leishmaniasis, and to avoid overtreatment and potential adverse events in patients having disease due to other etiologies.

Currently available diagnostic methods for CL have several limitations. Light microscopic analysis of smears obtained from cutaneous lesions is the most commonly used diagnostic method for CL because of its low cost, yet its sensitivity varies widely depending on the experience and skill of the operator. In addition, sensitivity of microscopy is diminished in lesions of longer duration in chronic lesions with their notorious low parasite burden [7,8]. The combination of lesion smear, culture of lesion aspirate or biopsy and histopathology (25%–50%

sensitivity) are diagnostic alternatives for chronic lesions [9], but their cost and restricted availability in reference centers, limit their use in rural settings. The Montenegro Skin Test is sensitive and specific but does not distinguish current from past infections [9–11], and since the requirement of production under standards of Good Manufacturing Practice, skin test antigen (leishmanin) is no longer available in the Americas.

A rapid test based on antigen detection for the diagnosis of CL at the point-of-care aimed principally at detection of *Leishmania major* and *Leishmania tropica* has been evaluated in the Old World, showing acceptable sensitivity (65%–68%) and specificity (80%–100%) [12,13]. Additionally, several molecular diagnostic tests have been developed for CL, achieving higher sensitivity (80% to 98%) and specificity (from 87% to 100%) than conventional diagnostic methods and establishing the feasibility of less invasive sampling [7,9,14–20].

Notwithstanding the challenges, the development of molecular tests for deployment where the disease occurs is a priority for improving access to care. Advances in non-invasive sampling for these molecular methods, such as lesion swabs, are particularly promising since they facilitate sample procurement for diagnosis in remote areas [15]. Nevertheless, the technical requirements and costs of sample processing by conventional or quantitative polymerase chain reaction (PCR) preclude their routine use in primary care facilities in resource-constrained settings.

Our team recently developed a method of Isothermal Recombinase Polymerase Amplification (RPA) targeting *Leishmania* kinetoplast DNA, coupled with a lateral flow (LF) immunochromatographic strip that has shown high accuracy in detecting *Leishmania Viannia* spp. [21]. This methodological approach allows sample processing and visual readout of results by naked eye using procedures amenable to local settings such as primary care centers. We reported the evaluation of the diagnostic performance of this RPA-LF test in a reference center laboratory, and in a field scenario where transmission is endemic, together with community participation.

## Methods

### Ethics statement

This research was approved and monitored by the Centro Internacional de Entrenamiento e Investigaciones Médicas (CIDEIM) Institutional Ethical Review Board (Approval number: 1275) in accordance with national and international regulations. Written Informed Consent was obtained from all participants or guardians of subjects <18 years of age. Assent was obtained from children ≥ 7 years of age. The study was registered under Clinical Trial Register NCT04500873. Medical decisions were based on the results of microscopic evaluation of lesion smear and culture, since at the time of the study, molecular methods were not yet recommended by the Colombian national guidelines for CL.

### Study design and population

We conducted a cross-sectional study of diagnostic test performance between January 2018 and July 2019. The study was carried out at CIDEIM in the reference laboratory in Cali, Valle del Cauca (3˚26'14.0"N 76˚31'21.0"W) and its primary health care facility in the municipality of Tumaco, Nariño, Colombia (1˚47'55.0"N 78˚48'56.0"W). CL is endemic in Tumaco but not in Cali, where suspected CL cases arriving from or who have visited endemic areas are referred for diagnosis and treatment. Participants ≥2 years of age, with ulcerated skin lesions of more than two weeks' duration were eligible and enrolled consecutively either in their residence (in rural Tumaco) by community health workers, or when seeking care at the primary health facility in urban Tumaco or the reference center laboratory in Cali. Sample size was calculated

using a formula to estimate a proportion [22,23]. One hundred fifteen participants were obtained with an expected 95% sensitivity for RPA-LF, 4% error and confidence interval of 95%. It was adjusted by 5% loss of enrollment patients, reaching a final sample size of 121 participants [24].

We evaluated the performance of the RPA-LF test to diagnose CL considering two scenarios. 1) Reference laboratory scenario: samples were obtained by highly trained technicians (auxiliary nurses) in a Primary Health facility in the urban area of Tumaco, then sent to the reference center in Cali where samples were processed by an expert microbiologist. This scenario also includes samples obtained from subjects seeking care directly at the reference lab in Cali (Fig 1A and S1 Fig). 2) Field scenario: samples were obtained in rural areas of Tumaco by trained community health workers (CHW), sent to and processed by a non-expert lab technician in the primary health facility in Tumaco who was trained by the research team to perform RPA-LF (Fig 1B and S1 Fig). The gold standard was constituted by the combination of several tests: lesion smear evaluated by microscopy, culture, histopathology (when the results of the

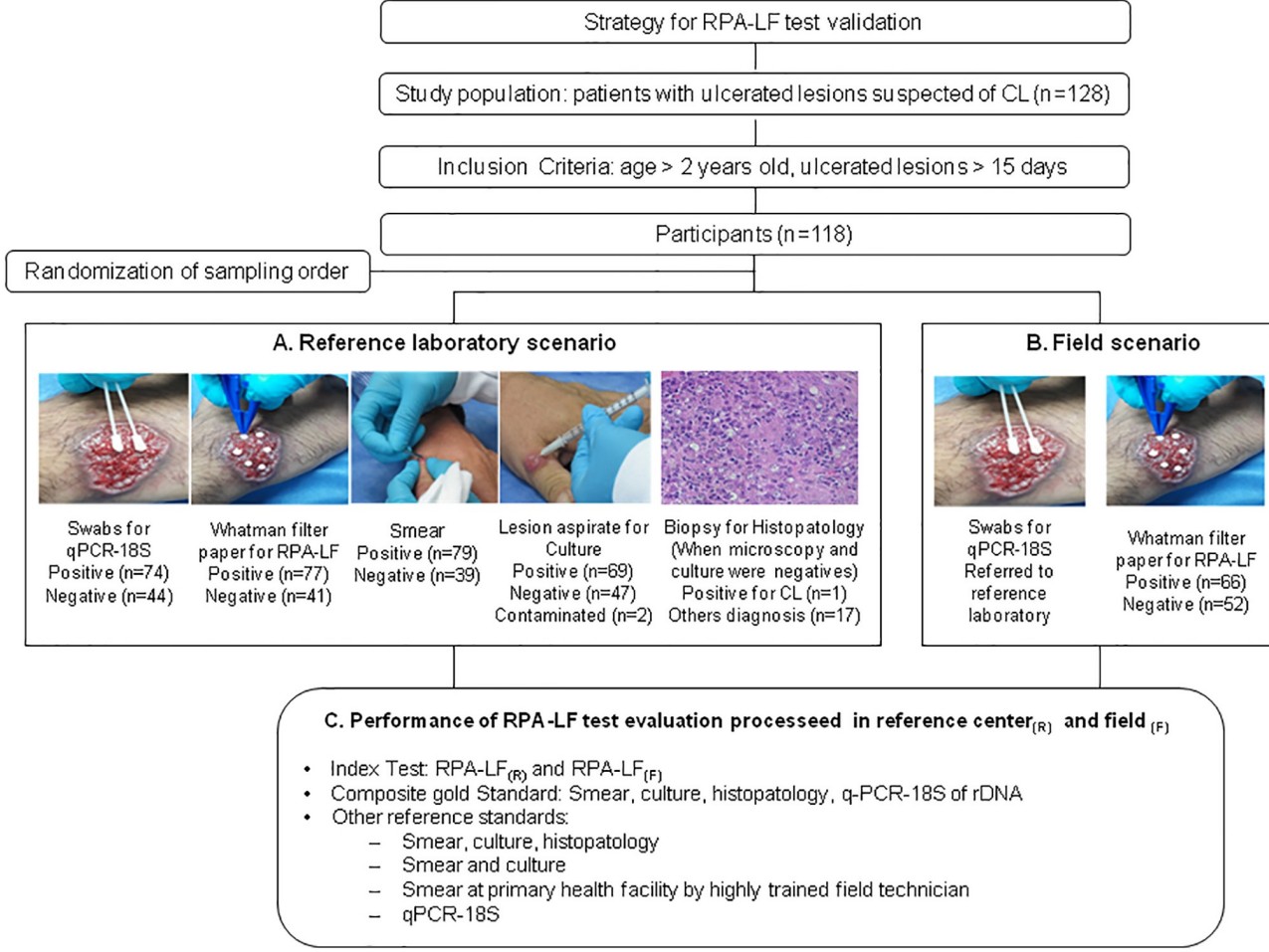

**Fig 1. Schematic summary of patient enrollment, and subsequent, sampling, diagnostic procedures, and performance of RPA-LF evaluation in two scenarios.** A) Reference lab scenario: samples were obtained by highly trained field technician in Tumaco and processed in a reference center in Cali. In Cali samples were obtained and processed by an expert microbiologist. B) Field scenario: samples were obtained by trained community health workers (CHW), RPA-LF was processed by a non-expert field technician in Tumaco in primary health facility (PHF), and swabs were sent to reference center to be processed. C. Performance of RPA-LF test.

two previously mentioned tests were negative), and real-time polymerase chain reaction detection of *Leishmania* 18S rDNA (qPCR-18S). The same composite gold standard was used to evaluate the performance of RPA-LF test in both previously described scenarios. Since this composite gold standard is not usually available outside the research settings, additionally secondary evaluations were done with non-reference standard tests (Fig 1C and S1 Fig).

### Data and clinical sample collection

Samples from enrolled patients were obtained by both the CHW at the place of residence in the rural area and by the trained technician at the primary health facility in the urban area of Tumaco, regardless of the site of initial enrollment.

**Reference laboratory scenario.**   The study physician collected clinical and demographic information by electronic data capture. Direct smear and culture, swabs and Whatman FTA filter papers from ulcerated lesions were obtained by highly trained field technician according with PAHO and other guidelines [21,25]. The order of these procedures was randomly allocated (block randomization). When smear and culture were negative, and based on physician assessment, a lesion biopsy was obtained and histopathologic evaluation was also performed for differential diagnosis. All swabs and Whatman filter paper samples obtained in the primary health facility in Tumaco were transported to the CIDEIM reference laboratory in Cali. Swab samples were stored at −20°C and Whatman filter paper discs at room temperature. Then they were processed by an expert microbiologist using q-PCR-18S of rDNA and RPA-LF testing, respectively (Fig 1).

**Field scenario.**   Ten CHW were selected to participate in the study according to the following criteria: being a member of and living in a rural community of the municipality of Tumaco, literacy, experience working in health, and acceptability by the community in accordance with WHO Guidelines for Community Health Worker Programmes [26,27]. CHW and two technicians of the primary health facility were trained using a structured program, with lectures and practical sessions. The training program included the following topics: ethics, basic concepts of CL and study procedures (research protocol, informed consent process, and collection, storage, and transportation of study samples). Additionally, field technicians were trained in the utilization of RPA-LF. Follow-up training at quarterly (3–4 month) intervals was carried out to re-enforce the technical capacity of the study team.

Samples were taken from the most recent ulcerated lesion in the following order: swabs for qPCR-18S and Whatman FTA filter paper for RPA-LF test. After cleaning and disinfecting the lesion and surrounding skin (iodized solution followed by 70% antiseptic alcohol and rinsing with sterile saline solution), two swab samples were obtained for qPCR-18S by gently rubbing over the surface of the ulcer ~10 times [15]. Afterwards, six 3 mm diameter Whatman FTA filter paper discs were applied to the lesion to absorb the tissue fluid and cells present on its surface. All samples (Whatman FTA filter paper and swabs) were stored and transported at room temperature to the local laboratory of the primary health facility in Tumaco where they were processed (Fig 1). Patients were also referred to this primary health facility for other laboratory tests within 7 days of the sampling procedure.

### Reference tests

**Composite gold standard.**   The "gold standard" for diagnosis of cutaneous leishmaniasis is visualization of amastigotes or isolation of *Leishmania*, yet individual methods to achieve this are not highly sensitive while molecular tests particularly those that amplify genetic material of *Leishmania* surpass the sensitivity of parasitological tests. For this reason, we utilized a composite "gold standard" based on microscopy of lesion smear, culture, histopathology of

biopsy and qPCR18S as a reference. Additionally, recognizing that at the point-of-care such a standard is not feasible, we also conducted comparative analyses of the performance of RPA-LF in relation with individual routinely used diagnostic tests: 1) microscopy of lesion smear, 2) direct smear and culture of aspirates 3) together with histopathology of biopsy and 4) qPCR-18S alone (Fig 1).

In accordance with the recommendations of the FDA for comparisons of new diagnostic tests with measures other than the "gold standard", comparisons with routine diagnostic procedures were made based on percent positive and negative agreement [28].

**Smear, culture and histopathology.** Microscopic evaluation of the standard direct smear was performed. A smear was considered positive when at least one intra- or extracellular amastigote was observed by microscopy. A result was negative when amastigotes were not observed in any of the examined fields. Four aspirates were obtained from the lesion border using a tuberculin syringe and 27G needle. Aspirates were cultured in Senekjie's diphasic culture medium [8] and parasite growth was evaluated at weekly intervals for up to one month. Histopathology was considered positive when amastigotes were visualized or the inflammatory pattern suggested leishmaniasis. All of these procedures were performed following the guidelines of PAHO [25]. *Leishmania* strains isolated from cutaneous lesions were identified using monoclonal antibodies [29]. Strains that were not identified with monoclonal antibodies, were analyzed by isoenzyme electrophoresis for species identification [30,31].

**DNA extraction and molecular amplification of *Leishmania* 18S rDNA.** DNA for qPCR of 18SrDNA was extracted from samples using Qiagen DNeasy Blood & Tissue Kit (Qiagen, USA) according to manufacturer's protocol. The resultant DNA was eluted in 50 μL AE buffer. Extraction controls without DNA samples were included to verify the absence of contamination during this process. The quantity and quality of nucleic acids was evaluated using a NanoDrop™ spectrophotometer. All DNA samples were stored at −20˚C until processing [15]. qPCR amplification of 18S r-DNA from *Leishmania spp*. was performed in a total volume of 12.5 μL (1.25 μL of total sample, 6.25 μL PCR Mastermix (BioRad), 1 μM of each oligonucleotide primer and 0.25 μM of the *Leishmania* 18S rDNA-specific FAM-labelled TaqMan probe and 2.75 μL Nuclease-Free Water [32]). The qPCR was carried out in the reference laboratory by an expert microbiologist. Comparisons between experiments were made using a standard curve for 18S r-DNA amplification of *L. (V) panamensis* DNA. Negative and positive controls were included in each PCR assay [15,33].

**Recombinase Polymerase Amplification Lateral Flow (RPA-LF).** The RPA-LF was conducted according to the methodology described by Saldarriaga and collaborators [21]. DNA extraction: 3 mm diameter disks were transferred to a 1.5 mL Eppendorf microcentrifuge tube using clean forceps, then washed three times for 5 minutes each at room temperature with 200 μL FTA Purification Reagent. Subsequently, the disks were washed once with 200 μL TE buffer (10 mM Tris-HCl, 1 mM EDTA) for 5 minutes at room temperature. After discarding the TE buffer, the disks were resuspended in 50 μL TE buffer and placed in a heat block at 95˚C for 30 minutes. The supernatant containing DNA was transferred to a prelabelled 1.5 mL microcentrifuge Eppendorf tube from which aliquots were drawn for molecular amplification. In this study we utilized the primer sequences and probe described by Saldarriaga et al (2016) [21]. The amplification mixture contained the forward primer-(5μM), biotinylated reverse primer (5μM), FAM-labeled probe (5μM), and the rehydrated cocktail (TwistAmp nfo RPA kit -TwistDx, UK). Two microliters of DNA extracted from the clinical sample and 1.25μL magnesium acetate 280nM were added to the mixture and amplified at 42˚C for 40 minutes using a dry bath. Then, 2μL of the RPA product were added to 98 μL of dilution buffer in a 1.5 Eppendorf tube and the lower edge of the lateral flow strip (Ustar Biotechonologies Hangzhou Ltda, China) was immersed in the solution. The amplification product migrated upwards by

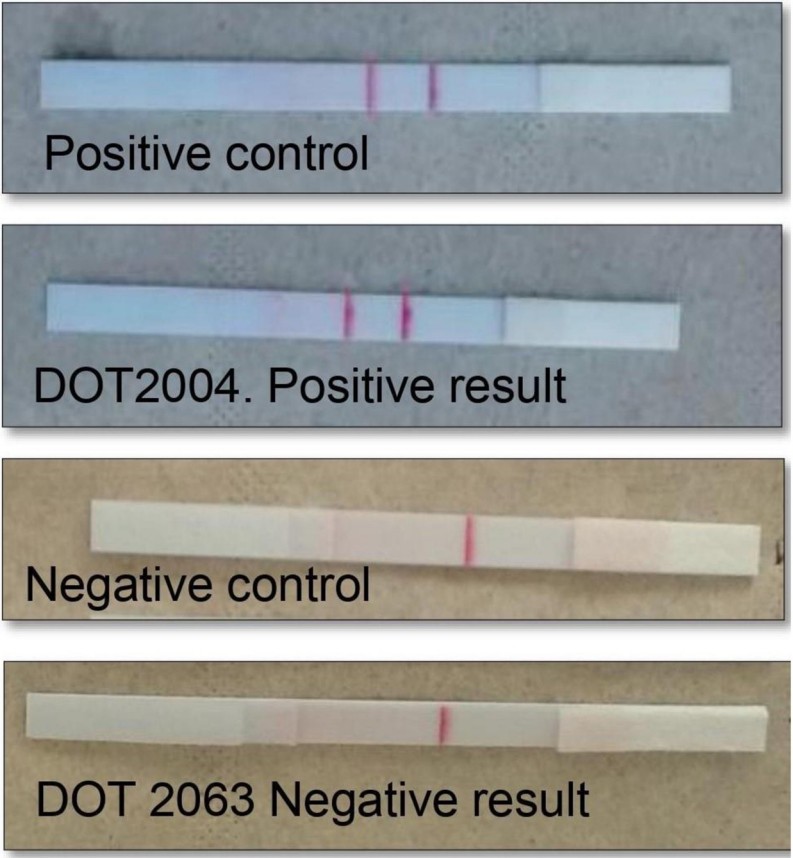

**Fig 2. Results of RPA-LF test**

capillarity. Parasite DNA amplification was visually confirmed within 10 minutes by the appearance of the corresponding 2 red bands in the lower portion of the strip (Fig 2). Negative and positive controls were included in each amplification batch [21]. The same procedure was used by the technician at the primary health facility in Tumaco and the expert microbiologist in the reference lab in Cali.

## Statistical analysis

We performed a descriptive analysis for clinical, socio-demographic and lesion characteristics. Acute CL was defined as individuals having lesions of <6 months of evolution and chronic those with lesions ≥ 6 months. We defined true positives based on at least one of the composite gold standard tests (smear, culture, histopathology or q-PCR-18S) and RPA-LF being positive. A result was considered true negative when all tests were negative. Similar definitions were used for each comparator Table 1.

When results evidenced contamination, double sequential RPA reactions were conducted using an aliquot of the first amplification for a second RPA. Samples of 29 participants, 25 from the primary health facility and 4 from the reference laboratory, were repeated due to contamination of the negative control, which invalidated the batch of samples. In this study, results were considered equivocal as defined by FDA guidelines [28].

We calculated sensitivity, specificity, predictive values (positive and negative) and likelihood ratios (positive and negative) with the corresponding 95% confidence intervals for both

**Table 1. Definitions of true positives and negatives according to reference tests.**

| Reference tests | True positive | True negative |
|---|---|---|
| Composite gold standard (smear, culture, histopathology and q-PCR-18S) vs. RPA-LF | At least one of the constituent tests of composite gold standard and RPA-LF were positive | All composite gold standard tests and RPA-LF were negatives |
| Smear vs. RPA-LF | Smear and RPA-LF tests were both positive | Smear and RPA-LF were both negative |
| Smear and culture vs. RPA-LF | At least one test and RPA-LF were positive | Smear and culture were negatives and RPA-LF was negative |
| Smear, culture, and histopathology vs. RPA-LF | At least one test and RPA-LF were positive | At least two reference tests and RPA-LF were negative |
| qPCR-18S vs. RPA-LF | q-PCR-18S and RPA-LF test were both positive | q-PCR-18S and RPA-LF were both negative |

scenarios, reference lab and field separately [34]. A McNemar test for paired data was used to identify differences in sensitivity and specificity and predictive values between reference lab and field scenarios [35–37]. Likelihood ratios were compared using the Differences in Diagnostic Likelihood Ratios Test, a P-value <0.05 was considered as statistically significant. Additionally, RPA-LF was compared with other test non- gold reference standard, estimating the positive and negative percent agreement with its confidence intervals in both scenarios following the FDA guideline [28]. Differences between reference laboratory and field were calculated using the McNemar test. Finally, sources of heterogeneity in the sensitivity of RPA-LF test compared with the composite gold standard were estimated using stratified analysis by sex, ethnicity, age, previous episode of leishmaniasis, medications received, number of lesions and duration of the oldest lesion. Stata, version 12 and DTCompair R software were used for these analyses.

## Results

### Study participants and gold standard results

After the initial enrollment of 128 CL suspected cases by the CHW, the physician at the primary care facility excluded ten participants due to lesion healing (n = 7) at the time of consultation or consent withdrawal (n = 3) (Fig 3 and S2 Fig).

Most participants were 18 years of age or older (78%), male (70.3%) and Afro-Colombians (62.7%). There was a wide range in the number of ulcers per patient (1–11), but single lesions were the most common presentation (53.3%). Most of the skin lesions (94.3%) were small ulcers (median: 15 mm diameter) localized on the arms (42%) or legs (37.6%). At the time of enrollment, the majority of lesions (89.3%) had less than six months of evolution. Only 5% of patients had a previous episode of CL as determined by the presence of typical scar or by clinical history Table 2 and S1 and S2 Tables.

The standard of care diagnostic protocol in Colombia is based on Giemsa-stained smears from skin lesions complemented by biopsy and histopathology when lesion smears are negative by microscopy. Seventy-nine patients presented positive smears, 69 were positive by culture and only one patient required confirmation by histopathology for a total of 80/118 (67.8%) patients being parasitologically confirmed. Additionally, q-PCR-18S identified three more cases, establishing that 83 (70.3%) of enrolled patients were positive for *Leishmania* spp. Among the 35 participants who were negative by the composite gold standard, 18 were diagnosed as having soft tissue infections by the treating physician hence biopsy and histopathology were not performed; the remaining 17 were evaluated by histopathology. The diagnoses

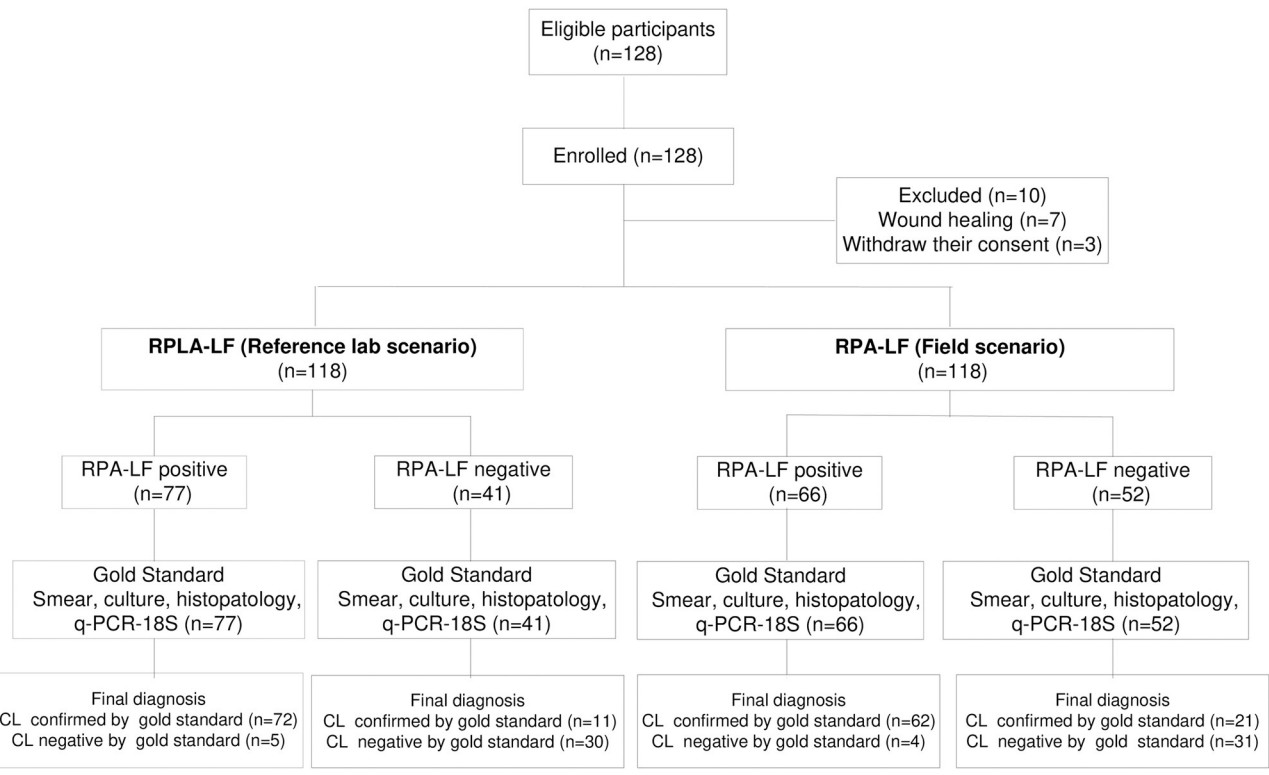

**Fig 3. Flow chart of participants by scenario.**

achieved by histopathology were vascular diseases (n = 12 71%), carcinoma (n = 3 17%) and soft tissue infections (n = 2 12%). Sixty-nine positive cultures allowed the identification of *L. (V.) panamensis* as the most frequent species (n = 64 94.8%) and the rest were *L. (V.) braziliensis* Table 2 and S1 Table.

## Performance of RPA-LF test

We determined that the RPA-LF test was capable of confirming or ruling out CL in both the lab reference center and field scenario using the composite gold standard as comparator. In the reference laboratory scenario, the sensitivity of RPA-LF was significantly higher at 87% (95%CI 79–94) than in the field scenario at 75% (95%CI 65–84), p = 0.04. This suggests that technical or logistic factors may have diminished test efficacy in the field. Specificity was similar in both scenarios: 86% (95%CI 74–97) and 89% (95%CI 78–99). Positive likelihood ratios were higher than 6, consequently RPA-LF is a good test to confirm CL in the reference center and field scenario. Likewise, negative likelihood ratios were between 0.2–0.3 showing that this test could be useful to discard CL Table 3 and S3 Table.

Comparison of the RPA-LF test with standards other than the composite gold standard that are used at the point-of-care to diagnose CL, reveal the positive percent agreement was significantly higher in the reference laboratory scenario than field scenario where agreement ranged between 85%–90% and 74–77% respectively. Negative percent agreement was lower than positive, and results for the reference laboratory and field were similar Table 4 and S4 Table.

**Table 2. Clinical and demographic characteristics of study participants.**

| Characteristics | n = 118 | |
|---|---|---|
| **Demographic** | | |
| Sex, n (%) | | |
| Male | 83 | (70.3) |
| Female | 35 | (29.7) |
| Ethnicity, n (%) | | |
| Afro-Colombian | 74 | (62.7) |
| Mestizo | 29 | (24.6) |
| Indigenous | 15 | (12.7) |
| Age, median (range), years | 26 | (2–85) |
| Department (state) of origin, n (%) | | |
| Nariño | 106 | (89.8) |
| Valle del Cauca | 7 | (6.0) |
| Others | 5 | (4.2) |
| **Clinical** | | |
| Previous episode of leishmaniasis, n (%) | 6 | (5.1) |
| *Leishmania* species, (n = 69), n (%) | | |
| L.(V) panamensis | 64 | (94.8) |
| L.(V) braziliensis | 5 | (7.2) |
| **Lesions (n = 245)** | | |
| Number of lesions. Median (range) | 1 | (1–11) |
| Duration of the oldest lesion, months, n(%) | | |
| 0–5.9 | 106 | (89.8) |
| ≥ 6 | 12 | (10.2) |
| Type of lesions, n (%) | | |
| Ulcer | 231 | (94.3) |
| Plaque | 10 | (4.1) |
| Other | 4 | (1.6) |
| Location in the body | | |
| Arms | 103 | (42) |
| Legs | 92 | (37.6) |
| Face-neck | 26 | (10.6) |
| Trunk | 24 | (9.8) |
| Presence of satellite lesions, n (%) | 33 | (13.5) |
| Presence of lymphadenopathy, n (%) | 13 | (5.3) |
| Maximum diameter of lesions (mm). Median (Range) (n = 236) | 18.6 | (3.2–78.4) |
| Maximum diameter of ulcers (mm). Median (Range) (n = 227) | 15 | (0.9–99.1) |

## Sources of variation, Compliance, and adverse events, of RPA-LF test

Sensitivity of RPA-LF in the reference laboratory was similar across of stratified analysis by sex, age, ethnicity, previous episode of leishmaniasis, medication received within the last month and, number of lesions. Most of variables had sensitivity variations between strata of less than 15%. The most relevant factor was duration of the lesion, which presented differences in sensitivity higher than 20%. Sensitivity of RPA-LF for lesions <6 months was 87.5% while lesions > 6 months, 66.7%. S5 Table.

All samples were obtained according to the protocol and no modifications were introduced to the RPA-LF test throughout the course of the study. The seven-day interval between

**Table 3. Diagnostic performance of RPA-LF in reference laboratory and field scenarios compared with composite gold standard (n = 118).**

| Scenario | TP | FP | FN | TN | Sensitivity % (95% CI) | p | Specificity % (95% CI) | p | PPV % (95% CI) | p | NPV % (95% CI) | p | LR+ % (95% CI) | p | LR- % (95% CI) | p |
|---|---|---|---|---|---|---|---|---|---|---|---|---|---|---|---|---|
| **Reference laboratory** | 72 | 5 | 11 | 30 | 87 (79–94) | 0.04* | 86 (74–97) | 0.7* | 94 (88–99) | 0.9£ | 73 (60–87) | 0.06£ | 6.1 (2.7–13.7) | 0.90 | 0.2 (0.1–0.3) | 0.7¶ |
| **Field** | 62 | 4 | 21 | 31 | 75 (65–84) | | 89 (78–99) | | 94 (88–100) | | 60 (46–73) | | 6.5 (2.6–16.6) | | 0.3 (0.2–0.42) | |

**PPV**: Positive predictive value. **NPV**: Negative predictive value. **LR**: Likelihood Ratio.

* McNemar Test.

£ Relative Predictive Values Test.

¶ Differences in Diagnostic Likelihood Ratio Test

**Table 4. Agreement between RPA-LF test and other diagnostic tests.**

| Diagnostic test | RPA-LF scenario | Positive percent agreement | | | Negative percent agreement | | |
|---|---|---|---|---|---|---|---|
| | | n/N | % (95%CI) | p* | n/N | % (95%CI) | p* |
| Smear | **Reference lab** | 67/71 | 94 (89–100) | 0.007 | 28/33 | 85 (72–97) | 0.3 |
| | **Field** | 55/71 | 77 (67–87) | | 25/33 | 75 (61–90) | |
| Smear + Culture | **Reference lab** | 72/79 | 91 (84–98) | 0.02 | 34/39 | 87 (76–98) | 1 |
| | **Field** | 61/79 | 77 (68–87) | | 34/39 | 87 (76–98) | |
| Smear + Culture + histopathology | **Reference lab** | 72/80 | 90 (83–96) | 0.04 | 33/38 | 86 (76–98) | 0.7 |
| | **Field** | 62/80 | 77 (68–87) | | 34/38 | 89 (79–99) | |
| qPCR-18S | **Reference lab** | 63/74 | 85 (77–93) | 0.08 | 30/44 | 68 (54–82) | 0.3 |
| | **Field** | 55/74 | 74 (64–85) | | 33/44 | 75 (62–88) | |

* McNemar Test

sampling by CHW and at the primary health facility was met in 91% of participants. None of the patients enrolled in the study presented adverse events related with the sampling procedures performed at the local clinic in Tumaco or by the community health workers in the rural areas.

## Discussion

This study evaluated the diagnostic performance of RPA-LF test in two scenarios, a primary health facility in a CL endemic area and a laboratory reference center. The test was compared against a composite gold standard that encompassed microscopy, culture, histopathology and qPCR. Use of a combination of highly sensitive diagnostic methods as gold standard sets a "high bar" for the comparative performance of RPA-LF, underrepresenting the benefit of this technology for endemic settings.

Results provided evidence that RPA-LF test is a valid test to confirm or rule out CL in a reference laboratory or field scenario. In a reference laboratory center the RPA-LF test accurately detected 87% of infections identified by the composite gold standard, which was the most stringent comparison. Sensitivity of the RPA-LF test was similar to other molecular tests that achieved values ranging from 81%– 98.7% [15,20].

This is the first study evaluating the performance of the RPA-LF test in which sampling was performed by CHW and processed in a primary health care "field scenario". Among the performance criteria, only the sensitivity of RPA-LF (75%) was significantly lower than the

reference laboratory scenario. The factors responsible for this difference still need to be determined, however we believe RPA-LF sensitivity could be improved by further simplification of the process, (e.g. reducing number of steps and materials involved).

During training sessions, we observed that manipulating the small filter paper discs (3 mm diameter) was challenging for the CHW. We consider that this difficulty could have affected, in part, the sensitivity of RPA-LF and that further improvement of the sampling protocol (i.e. by using swabs or larger filter paper discs) could optimize this point-of-care diagnostic test.

Differences in sensitivity between scenarios could also be attributed to the subjective reading of weak bands of the lateral flow strips when loaded with low concentrations of amplification products. Consequently, we believe that RPA-LF accuracy could be improved by adapting an inexpensive digital system that would add objectivity to the readout of results.

Some samples presented contamination. Yet, the areas of highest contamination risk are those found in labs where daily PCR work produce large numbers of amplicons. In the field, sampling with disposable materials offer lower risk of contamination. However, the local clinic should be aware of amplicons and establish separate areas for obtaining samples, extracting DNA and running the molecular test.

The simplicity of sample procurement using filter paper discs or swabs allowed trained CHW to obtain samples at the patients' homes, avoiding the need of costly transportation to the clinic, thereby overcoming a key barrier to diagnosis of CL. Furthermore, the implementation of noninvasive sampling is an advantage for all patients and particularly pediatric patients, who represented 9% of the study population and are even more frequent in other areas and settings.

Several studies have shown the importance of CHW participation in the diagnosis of other infectious diseases in areas with limited access to health services [38,39]. The benefit of our approach was that suspected patients were proactively identified by CHWs and samples were obtained and delivered to the local clinic for diagnosis. This strategy demonstrated its potential to reduce the health equity gap in hard-to-reach areas of transmission.

Regarding the comparison of RPA-LF with non-gold standard diagnostic tests that are used at the point-of-care to diagnose CL, we found a high (>85%) positive and negative agreement between them (smear, histopathology and culture in a research center). RPA-LF was capable of diagnosing CL in resource-limited settings, reaching or surpassing the individual and combined sensitivity of microscopy, culture and histopathology (≤80%), which are the most widely used diagnostic methods [9,18–20,40–42]. Additionally, the sensitivity of RPA-LF was higher than the sensitivity of lesion smears which is generally reported to be <60% [8,18,43–45].

Positive and negative agreements decreased when RPA-LF was compared to qPCR-18S, which is expected given that the qPCR is a real-time molecular test performed under ideal conditions. However, the RPA-LF test has practical advantages over qPCR-18S or other molecular methods because it is easier to perform, does not require expensive equipment, extensive training or sophisticated health infrastructure [14,40,46]. Other qualitative isothermal amplification methods, e.g. loop mediated amplification (LAMP), have shown a range of sensitivities for detecting *Leishmania* species [47–49]. Nevertheless, a drawback of qualitative LAMP is its dependency on indirect detection methods like turbidity or non-specific dyes that could potentially lead to false positive results [50].

Study strengths included randomization of the order in which samples were obtained avoiding potential bias that could have influenced RPA-LF performance. Results of RPA-LF were maintained across the different sources of variation such as sex, age, ethnicity, previous episode of leishmaniasis, number of lesions, ethnicity, age, receipt of medication, number of

lesions. Duration of disease was the most relevant variable that can modify the sensitivity of the test.

This study has some limitations. First, the RPA-LF process was not blinded and the operator could have known the smear results in advance, potentially influencing the interpretation of lateral flow reading. Second, not all patients with negative smear were evaluated by histopathology, and hence could have been misclassified as CL negative. Third, CL cases were mostly due to *L. (V.) panamensis*, consequently the efficacy of RPA-LF to detect other species within the subgenus *Viannia* requires additional evaluations. However, *L panamensis* and *L braziliensis* are the most relevant species in Colombia and the capacity of RPA-LF to detect other species of the subgenus *Viannia* has been previously determined [46]. People who migrated from other regions of Colombia may harbor other kinetoplastid parasites like *L. infantum*, *T. cruzi* or *T. rangeli* but the RPA-LF test does not cross-react with these parasites [21].

The sampling methods for RPA-LF or qPCR did not result in any adverse events. However, noninvasive samples can be obtained only from ulcerated lesions, limiting their usefulness in non-ulcerated and atypical forms of CL. It is possible that minimally invasive sampling using a fine needle (27G) aspirate and transfer of material to filter papers could overcome this limitation.

Recently, researchers affiliated with DNDi, FIND and WHO proposed the target product profile for a point-of-care CL diagnostic test as a "simple and robust test that can be implemented in resource-limited settings, enabling decentralized diagnosis and treatment of dermal leishmaniasis" [51]. Our results indicate that RPA-LF meets the principal characteristics of the product profile of a point-of-care test since they demonstrated its potential to improve access to CL diagnosis in resource-limited settings. Further studies should focus not only on test performance, but also on the operational requirements, costs, and relevant clinical outcomes. Utilization of this test should lead to a significant increase in the number of patients initiating treatment, and reduction of the time between appearance of disease and treatment [52–55].

In conclusion, RPA-LF is a valid, efficacious test to diagnose CL that could replace or complement microscopy in rural areas. Its combination with non-invasive sampling and low complexity requirements for processing and interpreting results position this test as an achievable alternative for diagnosis of CL at points of care in Colombia.

## Supporting information

**S1 Fig. Sampling and results of diagnostic procedures performed in reference laboratory and field scenarios.**
(CSV)

**S2 Fig. Results of diagnostic procedures, RPA-LF test in reference laboratory and field as well as results of composite gold standard.**
(CSV)

**S1 Table. Clinical and demographic characteristics of study participants.**
(CSV)

**S2 Table. Description of the study participants' lesions.**
(CSV)

**S3 Table. RPA-LF results in reference laboratory and field scenarios, as well as data of composite gold standard.**
(CSV)

**S4 Table. Results of diagnostic procedures, RPA-LF test in reference laboratory and field as well as results of composite gold standard.**
(CSV)

**S5 Table. Sensitivity of RPA-LF in the reference laboratory scenario stratified by sociodemographic and clinical characteristics.**
(DOC)

## Acknowledgments

We gratefully acknowledge the support of the CIDEIM Clinical Units in Tumaco and Cali, and the personnel of the Epidemiology and Biostatistics Unit, who prepared the databases and assisted in the analyses. We especially thank all the patients and Community Health Workers who participated in this study. We are grateful to the Hospital Divino Niño in Tumaco and Instituto Departamental de Salud de Nariño whose personnel supported community activities.

## Author Contributions

**Conceptualization:** Alexandra Cossio, Jimena Jojoa, María del Mar Castro, Lyda Osorio, Nancy Gore Saravia, Bruno L. Travi.

**Data curation:** Alexandra Cossio.

**Formal analysis:** Alexandra Cossio.

**Funding acquisition:** Alexandra Cossio.

**Investigation:** Alexandra Cossio, Jimena Jojoa, María del Mar Castro, Ruth Mabel Castillo, Lyda Osorio, Nancy Gore Saravia, Bruno L. Travi.

**Methodology:** Alexandra Cossio, Jimena Jojoa, María del Mar Castro, Ruth Mabel Castillo, Lyda Osorio, Thomas R. Shelite, Nancy Gore Saravia, Peter C. Melby, Bruno L. Travi.

**Project administration:** Alexandra Cossio.

**Resources:** Alexandra Cossio.

**Supervision:** Alexandra Cossio, Ruth Mabel Castillo, Nancy Gore Saravia.

**Validation:** Jimena Jojoa, Bruno L. Travi.

**Visualization:** Alexandra Cossio, Ruth Mabel Castillo, Nancy Gore Saravia, Bruno L. Travi.

**Writing – original draft:** Alexandra Cossio, Bruno L. Travi.

**Writing – review & editing:** Alexandra Cossio, Jimena Jojoa, María del Mar Castro, Ruth Mabel Castillo, Lyda Osorio, Thomas R. Shelite, Nancy Gore Saravia, Peter C. Melby, Bruno L. Travi.

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
