## [Decision Letter · Decision Letter 0]

8 Jan 2021

Dear Mrs. Cossio,

Thank you very much for submitting your manuscript "Diagnostic performance of a Recombinant Polymerase Amplification Test - Lateral Flow (RPA-LF) for cutaneous leishmaniasis in an endemic setting of Colombia" for consideration at PLOS Neglected Tropical Diseases. As with all papers reviewed by the journal, your manuscript was reviewed by members of the editorial board and by several independent reviewers. The reviewers appreciated the attention to an important topic. Based on the reviews, we are likely to accept this manuscript for publication, providing that you modify the manuscript according to the review recommendations. 

Sincerely,

Philippe Büscher, PhD

Associate Editor

Epco Hasker

Deputy Editor

Reviewer's Responses to Questions

**Key Review Criteria Required for Acceptance?**

**Methods**

-Are the objectives of the study clearly articulated with a clear testable hypothesis stated?

-Is the study design appropriate to address the stated objectives?

-Is the population clearly described and appropriate for the hypothesis being tested?

-Is the sample size sufficient to ensure adequate power to address the hypothesis being tested?

-Were correct statistical analysis used to support conclusions?

-Are there concerns about ethical or regulatory requirements being met?

Reviewer #1: (No Response)

Reviewer #2: This paper describes in details a relatively new technique of combining RPA and latter flow to develop a POC device, in this case for the detection of Leishmania.

The methodology of the study is very well described, and the graphic representation of the study design truly helps to grasp the context of the paper.

Just for the sake of clarity, a few additional details should be added, so that the paper becomes completely clear and will be used a a general refence by others in the future.

1) Line 277: please include primer sequences in the M&M section

2) Line 315 calculation methods (formulas) for sensitivity, specificity and positive / negative predictive should be added, taken that it is not always clear which golden standard is used exactly in the calculation. This reviewer was very pleased by the fact that PPV and NPV were include, something that is very often omitted in experimental diagnostic research papers despite the fact that these are the values that determine if a clinician can 'trust' the test in the field (while sensitivity and specificity are theoretic values that allow very easy masking of serious issues). So, well done ! 

The reason for this question: see below, result section.

Reviewer #3: lines 148-150 The authors could clarify for which hypothesis and power this sample size is calculated.

line 253 Are 'the samples' the swabs for qPCR?

**Results**

-Does the analysis presented match the analysis plan?

-Are the results clearly and completely presented?

-Are the figures (Tables, Images) of sufficient quality for clarity?

Reviewer #1: (No Response)

Reviewer #2: Very clear presentation of results. 

Just one question for additional explanation...

Line 369/370: the difference in 'results' obtained in the lab versus the field. Details are outlined in Table 3. 

From these results it is clear that the main differences are TP (72 vs 62) and FN (11 vs 21). of course it is obvious that field analysis will never perform as good as good as a lab context, but the question here is how did these 2 difference occur. By providing the formula used to calculate these values, and indicating which factor contributed to the decrease of TP and increase of FN, the work could become a textbook example of how to go about doing these sort of things. Again, the quality of the work, the methodology and paper is outstanding, and by adding this info, 'I' would even be temped to use this paper as a textbook/lecture example in advanced molecular parasitology classes.

Reviewer #3: lines 334-336 If the sample size was estimated to be 118, why did the authors include 128 patients?

**Conclusions**

-Are the conclusions supported by the data presented?

-Are the limitations of analysis clearly described?

-Do the authors discuss how these data can be helpful to advance our understanding of the topic under study?

-Is public health relevance addressed?

Reviewer #1: (No Response)

Reviewer #2: The conclusion is very well written, and this reviewer specially appreciated the critical assessment of the limitations of the study. Again, this is something that is very often missing from so many diagnostic papers, and it just adds to the value of this particular study. 

One last question however, RPA-LFA is not the cheapest technique, and by adding a 'cheap' readout tool in the future, the 'practicality' of the cost could further be an issue. Would it be possible for the authors just to add a few details on the current cost of the RPA-LFA (something that should be similar to RPA-LFAs for other diseases such as trypanosomosis) versus the cost of the 'current' most used golden standard in the field. the latter will be very different depending on the disease studied. It's not crucial for the quality of the paper, but it just allows the reader to think about the bigger picture and out everything in perspective.

Reviewer #3: line 433 - 443 Could sensitivity have been impacted by first doing swab sampling for qPCR and only then filter disks sampling for RPA-LF?

line 445 - 449 Is the misinterpretation of banding patterns also linked to the <90% specificity of RPA-LF in both field and ref lab?

**Editorial and Data Presentation Modifications?**

Reviewer #1: (No Response)

Reviewer #2: See other sections

Reviewer #3: line 59 remove the '('

line 63 duplicated 'in'

line 161 correct 'performed' to 'perform' 

line 351 duplicated 'in'

line 397 correct 'Ddiagnotic'

**Summary and General Comments**

Reviewer #1: The authors had previously established a RPA-LF protocol for CL diagnosis (Ref 21). Here, they successfully evaluate its efficiency and usefulness in field conditions by following a rigorous algorithm with multiple comparisons to an exhaustive panel of diagnostic tests (including a challenging composite gold-standard). This is an excellent work, with some real efforts to present the analytic pipelines and their corresponding results in a simple yet precise manner. The manuscript is well-written, the methodology is very rigorous and the discussion is adapted to the results. The results are positive and convincing and they will certainly be of interest for people involved in the control of CL, and possibly in the context of other NTDs for which diagnostic tools require improvement. I only have minor comments and suggestions below.

Minor comments:

• L.33: in the field?

• L.63: in both

• L.225: Leishmania in italic

• L.308-313: The relatively frequent occurrence of contaminations during the RPA protocol, especially in the field, could be explained a bit more in details and discussed in the appropriate section (possible reasons for such contaminations and ways to avoid it in the future).

Reviewer #2: Very clear paper.

Reviewer #3: (No Response)

PLOS authors have the option to publish the peer review history of their article (what does this mean?). If published, this will include your full peer review and any attached files.

Reviewer #1: Yes: Brice Rotureau

Reviewer #2: No

Reviewer #3: No
---

## [Decision Letter · Decision Letter 1]

5 Mar 2021

Dear Mrs. Cossio,

We are pleased to inform you that your manuscript 'Diagnostic performance of a Recombinant Polymerase Amplification Test - Lateral Flow (RPA-LF) for cutaneous leishmaniasis in an endemic setting of Colombia' has been provisionally accepted for publication in PLOS Neglected Tropical Diseases.

Best regards,

Philippe Büscher, PhD

Associate Editor

Epco Hasker

Deputy Editor

Reviewer's Responses to Questions

**Key Review Criteria Required for Acceptance?**

**Methods**

-Are the objectives of the study clearly articulated with a clear testable hypothesis stated?

-Is the study design appropriate to address the stated objectives?

-Is the population clearly described and appropriate for the hypothesis being tested?

-Is the sample size sufficient to ensure adequate power to address the hypothesis being tested?

-Were correct statistical analysis used to support conclusions?

-Are there concerns about ethical or regulatory requirements being met?

Reviewer #1: (No Response)

Reviewer #2: (No Response)

Reviewer #3: (No Response)

**Results**

-Does the analysis presented match the analysis plan?

-Are the results clearly and completely presented?

-Are the figures (Tables, Images) of sufficient quality for clarity?

Reviewer #1: (No Response)

Reviewer #2: (No Response)

Reviewer #3: (No Response)

**Conclusions**

-Are the conclusions supported by the data presented?

-Are the limitations of analysis clearly described?

-Do the authors discuss how these data can be helpful to advance our understanding of the topic under study?

-Is public health relevance addressed?

Reviewer #1: (No Response)

Reviewer #2: (No Response)

Reviewer #3: (No Response)

**Editorial and Data Presentation Modifications?**

Reviewer #1: (No Response)

Reviewer #2: (No Response)

Reviewer #3: (No Response)

**Summary and General Comments**

Reviewer #1: All suggestions and corrections were taken into account in the revised version.

Reviewer #2: Thank you for addressing the comments made during the first round of reviewing.

Reviewer #3: No further questions.

PLOS authors have the option to publish the peer review history of their article (what does this mean?). If published, this will include your full peer review and any attached files.

Reviewer #1: **Yes: **Brice Rotureau

Reviewer #2: No

Reviewer #3: No

---

## [Editor Report · Acceptance letter]

29 Mar 2021

Dear Mrs. Cossio,

We are delighted to inform you that your manuscript, "Diagnostic performance of a Recombinant Polymerase Amplification Test - Lateral Flow (RPA-LF) for cutaneous leishmaniasis in an endemic setting of Colombia," has been formally accepted for publication in PLOS Neglected Tropical Diseases.

Best regards,

Shaden Kamhawi

co-Editor-in-Chief

Paul Brindley

co-Editor-in-Chief
